# Consumption of a Fermented Milk Product Containing *Bifidobacterium lactis* CNCM I-2494 in Women Complaining of Minor Digestive Symptoms: Rapid Response Which Is Independent of Dietary Fibre Intake or Physical Activity

**DOI:** 10.3390/nu11010092

**Published:** 2019-01-04

**Authors:** Philippe Marteau, Boris Le Nevé, Laurent Quinquis, Caroline Pichon, Peter J. Whorwell, Denis Guyonnet

**Affiliations:** 1Sorbonne Université, INSERM, Laboratoire des Biomolécules (LBM), 27 rue de Chaligny, 75012 Paris, France; philippe.marteau@aphp.fr; 2APHP, Pôle Digestif, Hôpital Saint Antoine, 184 rue du Faubourg Saint-Antoine, 75012 Paris, France; 3Danone Nutricia Research, Innovation Science and Nutrition, 91767 Palaiseau Cedex, France; laurent.quinquis@danone.com (L.Q.); caroline.pichon@danone.com (C.P.); 4Wythenshawe Hospital, Manchester M23 9LT, UK; peter.whorwell@mft.nhs.uk; 5Diana Nova, 92110 Clichy La Garenne, France; dguyonnet@diana-group.com

**Keywords:** probiotic, *Bifidobacterium lactis* CNCM I-2494, gastrointestinal discomfort, digestive symptoms, physical activity, diet

## Abstract

**Background.** Minor digestive symptoms are common and dietary approaches such as probiotic administration or fibre and fermentable carbohydrate intake adjustments are often recommended. A Fermented Milk Product (FMP) containing *Bifidobacterium animalis* subsp. *lactis* CNCM I-2494 and lactic acid bacteria has been shown to improve digestive symptoms after 4 weeks of consumption, but the speed of onset of this effect and its dependence on fibre intake or physical activity is unknown. To answer these questions, data from two previously published trials on FMP for minor digestive symptoms were combined. **Methods.** In total, 538 participants provided weekly assessments of bloating, abdominal pain/discomfort, flatulence, borborygmi/rumbling stomach from which a composite score was calculated. At baseline in one study (*n* = 336), dietary fibre consumption was recorded and physical activity classified as high, moderate or low. The speed of the FMP’s effect was assessed by a repeated measure analysis of variance measuring the change from baseline for the composite score of digestive symptoms. **Results.** FMP consumption resulted in a significant decrease in the composite score of symptoms after only 2 weeks in both studies and the pooled data at week 1 (−0.35 [−0.69, 0.00]; *p* = 0.05), week 2 (−0.66 [−1.04, −0.27]; *p* < 0.001), week 3 (−0.49 [−0.89, −0.10]; *p* = 0.01) and week 4 (−0.46 [−0.88, −0.04]; *p* = 0.03). The interactions fibre intake-by-product group, physical activity-by-product group and time-by-product group were not statistically significant. **Conclusion.** FMP consumption leads to a rapid improvement in symptoms which is likely to encourage adherence to this dietary intervention. This effect is independent of dietary fibre and physical activity.

## 1. Introduction

Irritable bowel syndrome (IBS) as well as minor digestive symptoms place a significant burden on the health of the general population [1,2]. Dietary intervention is widely recommended to improve individual symptoms and well-being, especially reducing the consumption of fermentable oligo-di-mono-saccharides and polyols (FODMAPs), and supplementation with specific fibres (ex: psyllium) and some probiotics [3]. Physical activity seems to also have some positive effects [4]. Randomised controlled trials evaluating the efficacy of these approaches are limited and confounding factors are not always taken into account when evaluating the effect of a particular intervention. Our hypothesis was that lifestyle components such as fibre intake or physical exercise may influence the effect of dietary interventions on the improvement of abdominal discomfort. Furthermore, the speed of onset of the effect on gastrointestinal symptoms is likely to be a key factor in whether an individual continues to adhere to a dietary intervention, but it is seldom described.

Evidence that probiotics improve digestive discomfort is presently limited to a relatively limited number of products [3,5,6]. The most studied preparations contain bifidobacteria, but only some specific strains have demonstrated beneficial properties [7]. Capsules containing *Bifidobacterium infantis* 35,624 have been shown to improve symptoms in patients with IBS [7] and the absence of efficacy in 275 subjects without diagnosed gastrointestinal disorder complaining of discomfort might be due to differences in the population studied [8]. A Fermented Milk Product (FMP) containing *Bifidobacterium lactis* CNCM I-2494 and lactic acid bacteria has been shown to improve IBS symptoms [9,10]. It also improved gastrointestinal well-being and symptoms of discomfort in subjects from the general population with minor digestive symptoms [11,12]. However, the speed of improvement (i.e., effect before 4 weeks of consumption) and the potential interaction with fibre intake and physical activity have not been evaluated.

This work aimed to explore the available data from two previously published studies to determine the speed of improvement of abdominal discomfort with a FMP containing *B. lactis* CNCM I-2494 and to establish whether lifestyle components such as fibre intake or physical exercise may influence this effect.

## 2. Materials and Methods

The datasets analysed in this research were obtained during two double-blind randomised controlled trials using the same design, products and endpoint derivation [11,12]. The primary endpoint, which was the effect of the consumption of a FMP over 4 weeks on Subject’s Global Assessment (SGA) of gastrointestinal well-being, was met in study 1 and on pooled data from the two trials, but not on study 2. The secondary endpoint which was the effect of the consumption of a FMP over 4 weeks on the composite score of digestive symptoms was met in both trials. Briefly, 538 subjects were randomised through advertisement, 202 in study 1 [11] and 336 in study 2 [12] with the same inclusion and exclusion criteria. These were adult women from the general population, aged between 18 and 60 years with a Body Mass Index (BMI) range of 18–30 kg/m^2^, complaining of minor abdominal discomfort and with a bowel movement frequency within 3–21 per week (the normal range). Subjects were excluded if they had a clinical diagnosis of any digestive disease including IBS, or they received drugs for digestive symptoms, or they had any significant systemic disease. Subjects were not allowed to consume any other probiotic than the study products and were asked to not change their usual dietary intake and or level of physical activity. The consumption of fibre was measured at baseline during a phone interview with a dietician in study 2 (*n* = 336) and subjects were divided into four levels corresponding to quartiles of fibre consumption. Physical activity was assessed at baseline in study 2 using the international physical activity questionnaire (IPAQ) which allows the classification of subjects into three categories of physical activity (high, moderate, and low) [13].

Subjects self-evaluated weekly the frequency of four individual digestive symptoms (abdominal pain/discomfort, bloating, flatulence/passage of gas and borborygmi/rumbling stomach) using a 5-point Likert scale. A composite score of discomfort ranging from 0 to 16 was calculated as the sum of those four symptom scores. After a 2-week run-in period, subjects were randomised in a 1:1 ratio to consume 125 g of either a non-fermented dairy product without bacterial strains (control product) or a FMP containing *Bifidobacterium animalis* subsp. *lactis* CNCM I-2494, *Streptococcus salivarius* subsp. *thermophilus* CNCM I-1630, *Lactobacillus delbrueckii* subsp. *bulgaricus* CNCM I-1632 and I-1519, and *Lactococcus lactis* subsp. *lactis* CNCM I-1631 twice a day for 4 weeks. Both products were prepared at Danone Research facilities, Palaiseau, France, and shipped in blinded packaging with refrigeration to the study sites. The randomisation was generated in well-balanced blocks using envelopes for study 1 by an external contract research organization (LC2) and using Interactive Web Response Systems for study 2 developed by another external contract research organization (Axonal). Blinding was accomplished by ensuring that test and control products were of identical appearance, taste and texture. The randomisation code was not to be broken until all assessments had been performed, all data had been entered into the database, and the database had been locked after a clean file procedure.

## 3. Statistical Analyses

Statistical analyses were performed using mixed procedure available in the SAS System package (SAS Institute Inc., Cary, NC, USA) version 9.3. All the analyses were performed on the intention to treat (ITT) principle based on all randomised subjects. No imputation techniques were used to handle for missing data on endpoint assessment.

To assess the FMP’s effect on the frequency of digestive symptoms, a secondary endpoint in both studies, we used a repeated-measure analysis of variance (ANOVA) on the change from baseline with fixed effect of study, time (in week) and product group, as well as time-by-product group interaction effect. Additional covariates were included such as baseline digestive symptoms composite score and study-by-product group interaction (Model 1 on pooled data).

For exploratory purposes, the Least Squares (LS)-Means and their 95% Confidence Interval (CI) were produced by study product group and by study product group per time. Two-sided approximate *t*-test *p*-values comparing the LS-Means difference between study product groups were estimated globally and by time point. Although this analysis was exploratory, we performed as multiple comparisons the hierarchical stepdown multiple-testing strategy starting from timepoint Week 4 (W4) and going back week-by-week until Week 1 (W1). The results are graphically displayed with series plots showing LS-Means difference (FMP-Control) with 95% CI.

For the dietary fibre consumption and physical activity impact on the change from baseline of digestive symptoms and the interaction with study product group, we added separately in two repeated-measure ANOVA the qualitative covariate fibre or physical activity as well as the interaction fibre—or physical activity-by-time and fibre—or physical activity-by-product group (Model 2 and Model 3 on Study 2 data, respectively). An interaction type 3 test of fixed effect *p*-value lower than 0.1 was used as the threshold to display the LS-Means and their 95% CI by interaction groups.

## 4. Results

The analyses to assess the speed of the FMP’s effect on abdominal discomfort were performed on the pooled dataset representing 538 subjects (pooled ITT population). The number of subjects with missing data for the frequency of digestive symptoms was too low (3 out of 538; 0.6%) to use any missing data strategy for replacement. Dietary intake of fibre and Physical Activity (PA) data were only gathered in Study 2, so we performed the analyses of the impact of fibre intake and physical activity on response to FMP consumption on a set of 336 subjects (study 2 ITT population). The number of missing data for fibre intake and physical activity was 28 (8.3%) and 39 (11.6%), respectively. The model was run with the subjects having available information (308 subjects for dietary fibre intake and 297 subjects for physical activity).

### 4.1. Characteristics of the Subjects and Groups

The subjects’ characteristics are shown in Table 1 and Table 2. The daily fibre intake at baseline was balanced between the FMP and Control group (15.3 and 15.6 g/day respectively) and the distribution in quartiles is shown in Table 2. The subjects’ physical activity (PA) was high, moderate or low in 116 (34.5%), 138 (41.7%), and 43 (12.8%) respectively and was consistent between study product groups with 54 (32.1%), 75 (44.6%) and 19 (11.3%) in the FMP group and 62 (36.9%), 63 (37.5%) and 24 (14.3%) in the Control group. Digestive symptoms at baseline were homogeneous in the 3 groups of PA.

### 4.2. Speed of the FMP’s Effect on Abdominal Discomfort

FMP led to a significantly greater reduction of the LS-Means difference of Composite Score (CS) over the 4-week consumption period as compared to the Control product (−0.49 [−0.82, −0.16]; *p*-value = 0.003). This greater reduction of CS by FMP was already present and significant after 2 weeks of product consumption (Figure 1). The LS-Means differences FMP–Control of CS and their [95% CI] across the study time points (Week 1, Week 2, Week 3, Week 4) were always in favour of the FMP (Table 3).

### 4.3. Impact of Baseline Levels of Fibre Intake and Physical Activity on Intervention Effect

The greater reduction observed for FMP as compared to the Control of the Composite Score (CS) over the 4-week consumption period was not affected by the adjustment for baseline levels of fibre intake and physical activity (LS-Means and 95% CI equals to −0.36 [−0.75; 0.04]; *p*-value = 0.08 and −0.48 [−0.96; 0.00]; *p*-value = 0.05, respectively) (Table 3). There was no significant interaction effect of fibre intake (*p* = 0.25) or physical activity (*p* = 0.75) with the study product.

### 4.4. Safety

In the two studies, no adverse events or serious adverse events were reported as being related to the study product. Overall clinical safety was good.

## 5. Discussion

This research indicates that consumption of a FMP over 4 weeks can improve digestive symptoms after only 2 weeks of product consumption. Furthermore, the data suggest that this effect is not influenced by dietary fibre intake or physical activity at the start of study.

Abdominal discomfort and irritable bowel syndrome (IBS) are common disorders with the latter being characterised by similar but more intense symptoms on a chronic basis. The pathophysiology of IBS is multifactorial and includes disturbances of gastrointestinal motility and sensitivity as well as psychological influences. In addition, there is now evidence that the gut microbiota is disturbed (dysbiosis) in at least some patients with the disorder.^14^ Furthermore, the observation that the intestinal and central aspects of the disorder can be transmitted by faecal transplantation in animal models and potentially modulated by probiotics, especially bifidobacteria, opens space for the development of new treatments [5,6,14].

In the present study, we used data collected in two randomised controlled studies which had shown that a FMP was more effective than a non-fermented control product in improving discomfort over a 4-week period of consumption, which is the time frame recommended in guidelines for trials in functional gastrointestinal disorders. We considered it to be of interest to explore how fast this effect occurred as it might be anticipated that the absence of measurable effect after 2 weeks could result in poor compliance or even the cessation of consumption by participants. We found that the FMP had a rapid effect and interestingly a similarly fast response has been observed in other treatments aimed at modifying the microbiota, such as rifaximin [15].

It is noteworthy that the FMP containing *B. lactis* CNCM I-2494 has been shown to modulate intestinal physiology in healthy subjects [10,16] and improve symptoms in both patients with IBS^9,10^ as well as those with minor digestive symptoms [11,12]. Conversely, capsules containing the probiotic *B. infantis* 35624 improved symptom scores in patients with IBS but not in subjects without diagnosed gastrointestinal disorder with less severe symptoms [7,8,17]. However, it should not be assumed that all lactic acid bacteria share the same physiological and therapeutic effects. In fact, worsening of IBS symptoms has been reported with some products [18] and others appear to have differential effects on abdominal discomfort and associated psychological behaviours [19].

Fibre intake can modulate digestive symptoms and lowering the consumption of fibre [20] as well as fermentable carbohydrates (FODMAPs) is recommended in IBS [3]. Furthermore, exercise can have an effect on the gut and this is particularly noticeable in athletes [21,22]. We did not find any evidence of a significant interaction between baseline levels of fibre intake and physical activity and the effect of the FMP in our population. However, it has to be acknowledged that the effect of fibre intake on Gastrointestinal (GI) well-being is complex and, for instance, we did not assess the ingestion of specific types of fibre. Some types of fibre, especially the insoluble ones, increase faecal volume and may help to relieve constipation [23]. Conversely, the excessive consumption of soluble fibre which is more fermentable, may increase symptoms of bloating and excessive gaseousness in individuals with gastrointestinal hypersensitivity. However, our findings do not suggest that the beneficial effect of the FMP on GI symptoms is dependent on the intake of dietary fibre.

The present research has some limitations. Firstly, we do not have information on the effect of the FMP consumption on a long-term basis and it would be useful to know whether improvement is sustained after discontinuation of treatment or whether continued consumption is necessary. Secondly, this was an exploratory post-hoc analysis based on secondary endpoints from previously published studies. Thirdly, neither study, alone or combined, was prospectively designed to confirm the absence of an interaction between the product effect and level of physical activity or fibre intake at baseline. As such, the associated interactions tests may lack statistical power. Lastly, there was no assessment of the effect on the gut microbiota or its metabolites which might improve understanding of the mechanisms behind the observed clinical effect.

In conclusion, early phase studies of potential new products for minor digestive symptoms should probably assess the speed of onset of any beneficial effect, as this is likely to have a major impact on whether it is likely to be widely used in this setting. In addition, it would be useful to know whether response to dietary intervention is maintained after discontinuation of the intervention or whether maintenance consumption is necessary. However, the feasibility of such long-term studies with a randomised controlled design is questionable. Real-world evidence studies might be an option in this context.

## Figures and Tables

**Figure 1 nutrients-11-00092-f001:**
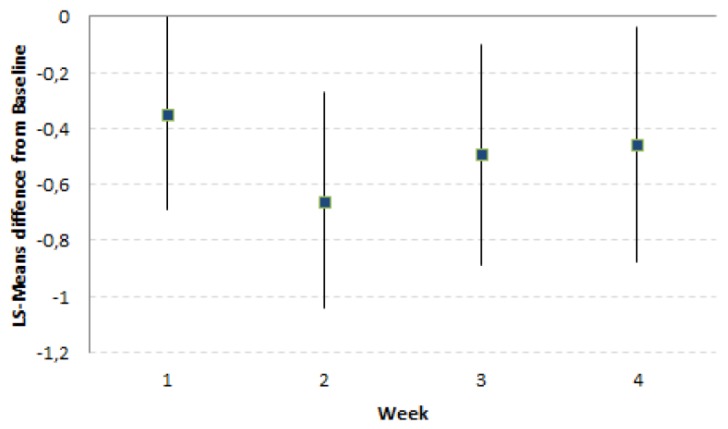
LS-Means differences and 95% confidence interval (CI) (FMP–Control) from Baseline of composite score of digestive symptoms by week (*N* = 535).

**Table 1 nutrients-11-00092-t001:** Baseline characteristics of subjects.

	Study 1 (*N* = 202)		Study 2 (*N* = 336)		Pooled (*N* = 538)	
	FMP (*N* = 102)	Control (*N* = 100)	FMP (*N* = 168)	Control (*N* = 168)	FMP (*N* = 270)	Control (*N* = 268)
Age (years)	31.6 ± 9.5	31.8 ± 10.4	32.1 ± 10.2	32.9 ± 10.9	31.1 ± 9.9	32.5 ± 10.7
BMI (kg m^−2^)	23.2 ± 2.7	23.2 ± 2.6	22.9 ± 2.5	23.0 ± 2.9	23.0 ± 2.5	23.1 ± 2.8
Composite score of digestive symptoms *	7.0 ± 2.2	7.2 ± 2.3	6.9 ± 2.2	6.9 ± 2.1	6.9 ± 2.2	7.0 ± 2.2
Fibre intake ^+^ (g/day)			15.3 (5.0)	15.6 (5.1)		
Physical activity ^§^						
*Missing*			19 (11.3%)	20 (11.9%)		
*Low*			24 (14.3%)	19 (11.3%)		
*Moderate*			63 (37.5%)	75 (44.6%)		
*High*			62 (36.9%)	54 (32.1%)		

FMP—Fermented Milk Product. All data are expressed as mean ± SD except ^§^ physical activity expressed as percentage and number of subjects by class [*n* (%)]. * Composite score of frequency of digestive symptoms ranged from 0 to 16. ^+^ The statistics are displayed for subjects with data on fibre intake (308 out of 336; 91.7%).

**Table 2 nutrients-11-00092-t002:** Baseline characteristics of subgroups of subjects in Study 2 according to fibre intake and physical activity.

	Age (years)	BMI (kg m^−2^)	Composite Score
Fibre intake *			
Q1 (*N* = 77)	32.9 ± 10.6	23.3 ± 2.7	6.8 ± 2.3
Q2 (*N* = 77)	32.7 ± 10.9	22.6 ± 2.3	6.8 ± 2.1
Q3 (*N* = 77)	30.6 ± 10.6	22.7 ± 2.6	7.1 ± 2.0
Q4 (*N* = 77)	34.8 ± 10.1	23.1 ± 3.0	6.8 ± 2.2
Physical activity			
Low (*N* = 43)	34.2 ± 11.3	23.0 ± 3.0	6.6 ± 2.2
Moderate (*N* = 138)	32.6 ± 10.2	22.9 ± 2.6	7.0 ± 2.2
High (*N* = 116)	32.7 ± 9.5	22.7 ± 2.5	6.9 ± 2.0

All data are expressed as mean ± SD. * The statistics are displayed for subjects with data on fibre intake (308 out of 336; 91.7 %) and on physical activity (297 out of 336; 88.4%).

**Table 3 nutrients-11-00092-t003:** LS-Means differences and their 95% confidence interval over 4 weeks and by week.

	Week 1	Week 2	Week 3	Week 4	Over 4 Weeks
**Model 1:** **Pooled data** ***N* = 535**	−1.01; −0.66 −0.35 [−0.69, 0.00] 0.05	−1.53; −0.87 −0.66 [−1.04, −0.27] <0.001	−1.82; −1.33 −0.49 [−0.89, −0.10] 0.01	−2.07; −1.61 −0.46 [−0.88, −0.04] 0.03	−1.61; −1.12 −0.49 [−0.82; −0.16] 0.003
**Model 2:** **Fibre data** ***N* = 308**	−0.72; −0.29 −0.43 [−0.82; −0.03] 0.04	−0.90; −0.43 −0.48 [−0.93; −0.02] 0.02	−1.23; −0.95 −0.28 [−0.77; −0.2] 0.25	−1.46; −1.22 −0.24 [−0;76; −0.28] 0.37	−1.08; −0.72 −0.36 [−0.75; 0.04] 0.08
**Model 3:** **PA data** ***N* = 297**	−0.60; −0.27 −0.33 [−0.82; 0.17] 0.19	−0.82; −0.23 −0.60 [−1.12; −0.07] 0.03	−1.35; −0.88 −0.47 [−1.02; 0.09] 0.10	−1.67; −1.16 −0.51 [−1.10; 0.07] 0.08	−1.11; −0.63 −0.48 [−0.96; 0.00] 0.05

Results are expressed in each row as (i) LS-Means change from baseline in respectively FMP and Control group, (ii) difference (FMP–Control) in LS-Means difference (iii) 95% confidence interval for this difference and (iv) two-sided approximate *t*-test *p*-value comparing the LS-Means differences to 0. **Model 1:** Estimated via a repeated linear mixed model on change from baseline on the composite score with fixed effect of product group, time, study, with covariates baseline composite score as well as the interactions time-by-product group and study-by-product group. **Model 2:** Estimated via a repeated linear mixed model on change from baseline on the composite score with fixed effect of product group, time, with covariates baseline composite score and fibre intake, as well as the interactions time-by-product group, time-by-fibre intake and fibre intake-by-product group. **Model 3:** Estimated via a repeated linear mixed model on change from baseline on the composite score with fixed effect of product group, time, with covariates baseline composite score and physical activity, as well as the interactions time-by-product group, time-by-physical activity and physical activity-by-product group.

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
