# Peer review of "Consumption of a Fermented Milk Product Containing Bifidobacterium lactis CNCM I-2494 in Women Complaining of Minor Digestive Symptoms: Rapid Response Which Is Independent of Dietary Fibre Intake or Physical Activity"

_nutrients, 2019, doi:10.3390/nu11010092_

Reviewer 1 Report

This is an interesting study. Comments are:

The introduction needs a clearer presentation of hypotheses that lead to test these complicated models.

In the results, I believe it wrong to present Table 3 first. This definitely needs rearrangement. It took me some time to decide what it actually meant. I also think it needs much more of a discussion what these numbers actually mean. Linear repeated models always need more explanation. Also, why interactions where chosen.

Next, model diagnostics have not been presented. Can you kindly so?

Graph four needs to be clearer if possible. Can table 3 be converted to a graph possibly?

Author Response

Reviewer#1
This is an interesting study. Comments are:

The introduction needs a clearer presentation of hypotheses that lead to test these complicated models.

We added the following sentence to the introduction: “Our hypothesis was that life style components such as fibre intake or physical exercise may influence the effect of dietary interventions on improvement of abdominal discomfort.” Hoping this is now clearer.

In the results, I believe it wrong to present Table 3 first. This definitely needs rearrangement. It took me some time to decide what it actually meant.

We agree. We propose to move Table 3 to results section “Speed of FMP’s effect on abdominal discomfort”. We hope that the flow is now clearer to the reader.

 I also think it needs much more of a discussion what these numbers actually mean. Linear repeated models always need more explanation.

Numbers in each cell of Table 3 are LS-Means change from baseline for FMP and Control followed by the difference (FMP-Control) in LS-Means and 95% Confidence Interval (CI) for this difference. A negative LS-Mean difference represents a beneficial effect of FMP over Control and a positive LS-Mean difference represents a beneficial effect of Control over FMP.

The LS-Means, corresponding differences and CIs were estimated using a repeated linear mixed model with fixed effect of product group (categorical variable, 2 groups) and time (categorical variable, 4-time-points in total: Week 1, Week 2, Week 3, Week 4), as well as time-by-product group interaction effect.  Additional terms were included for baseline digestive symptoms composite score as a covariate, study as a fixed effect as well as the effect of study-by-product group interaction in Model 1. Fibre intake at baseline was included as a covariate in Model 2. Physical activity was included as a covariate in Model 3. Subject was fitted as a random effect and an unstructured covariance matrix was specified.

Also, why interactions where chosen.

In Model 1 we used the interaction time-by-product group to allow product differences to be estimated for each Week. The interaction study-by-product group was used to test if the effect on Composite Score change was homogenous between the two studies.

For Model 2 and Model 3 additional covariates-by-time and covariates-by-study product with respectively fibre intake and physical activity were used to address if the level of fibre intake and physical activity at baseline had a differential effect on the dependent variable.

Next, model diagnostics have not been presented. Can you kindly so?

We assume you are referring to the ANOVA model assumptions of residuals normally distributed with equal variance. We checked this using a Normal probability plot of the studentized residuals versus expected Normal scores (see figure below for Model 2).     

Additionally, to calculate degrees of freedom in the repeated model ANOVA we used the Kenward Roger method approximation. The structure of variance covariance used was the unstructured one. We did not evaluate the impact of a different type of matrix structure for this model.  

Graph four needs to be clearer if possible. Can table 3 be converted to a graph possibly?
We assume you are referring to Figure 1 when mentioning “graph four”. As requested, we converted Table 3 data to a graph (see below New Figure 1 with 3 panels corresponding to the 3 models). In case Table 3 is to be replaced by New Figure 1 in the manuscript, we propose to move Table 3 in supplementary material. In this case, former Figure 1 would be of no further use.

New Figure 1: LS Means difference (FMP-Control) on change from baseline in composite score of digestive symptoms

New Figure 1 legend:

Model 1: Estimated via a repeated linear mixed model on change from baseline on the Composite Score with fixed effect of product group, time, study, with covariates baseline Composite Score as well as the interactions time-by-product group and study-by-product group.

Model 2: Estimated via a repeated linear mixed model on change from baseline on the Composite Score with fixed effect of product group, time, with covariates baseline Composite Score and fibre intake, as well as the interactions time-by-product group, time-by-fibre intake and fibre intake-by-product group.

Model 3: Estimated via a repeated linear mixed model on change from baseline on the Composite Score with fixed effect of product group, time, with covariates baseline Composite Score and physical activity, as well as the interactions time-by-product group, time-by-physical activity and physical activity-by-product group.

The same figure can be found below with the addition of 95%CI values of the LS-Means difference by visit. We personally find it slightly more difficult to read, but we let the final decision to the reviewers and editors.

Reviewer 2 Report

This is an interesting sponsored study.

 Comments:

The difference between minor digestive disorders and IBS is not very clear. Do you have Rome III criteria? Can you specify more?

If we look at references 11 and 12, the name of the product differs (DN-173 010 and CNCM I-2494): please explain

The evaluation pooled on a secondary endpoint poses a methodological problem making questionable the interpretation of the study even if this is discussed by the authors.

Minor

Introduction: Specify what you mean by "specific fibers"

Specify certain abbreviations: CRO (LC2), LS

The tables are not in the correct order in the article

Author Response

Reviewer#2

This is an interesting sponsored study. Comments:

The difference between minor digestive disorders and IBS is not very clear. Do you have Rome III criteria? Can you specify more?

We acknowledge that we did not detail in the present manuscript the inclusion and exclusion criteria used in the two original RCTs (Guyonnet D et al, 2009; Marteau P et al, 2013).

Only subjects having a mean composite score between 2 and 12 [score ranging from 0 (no symptom) to 16 (all symptoms every day)] during a 2-week run-in period and meeting the other randomization criteria (normal bowel movement frequency, no consumption of antibiotics) were randomized to the intervention.

Subjects were excluded if: they were fulfilling criteria of any functional gastrointestinal disorder (including IBS) according to ROME III criteria, they already consulted a gastroenterologist or general practitioner for digestive symptoms of the lower tract, they took or were under prescription of treatment for digestive symptoms (e.g. antispasmodic, laxatives or antidiarrheal drugs), and they had any significant systemic disease. Antibiotic ingestion within the month prior to the entry in the study was also an exclusion criterion.

Individuals with known lactose intolerance or with dietary habits which might interfere with the assessment of the study product (e.g. slimming or vegetarian diets) or known allergy to the study product components were also excluded. Throughout the study, the subjects were not allowed to consume any probiotic (including food supplements) or fermented dairy product other than those provided. They were encouraged to continue with all the other aspects of their dietary and physical exercise habits.

If we look at references 11 and 12, the name of the product differs (DN-173 010 and CNCM I-2494): please explain

Agreed and we realize this can be confusing for the reader. Some past publications used the internal Danone collection ID “DN-173 010” to identify the bifidobacterium strain also referenced as CNCM I-2494 in the “Collection Nationale de Cultures de Microorganismes” at Pasteur Institute, Paris. This is the same bacterial strain.

The evaluation pooled on a secondary endpoint poses a methodological problem making questionable the interpretation of the study even if this is discussed by the authors.

As stated by the reviewer, we acknowledge this limitation in the discussion. However, we consider that results from the present exploratory work may be useful when designing future studies and deserve therefore attention.

Minor

Introduction: Specify what you mean by "specific fibres"

An example of fibre commonly used to treat symptoms of constipation (psyllium) has been provided to illustrate this sentence in introduction.

Specify certain abbreviations: CRO (LC2), LS

CRO stands for “contract research organization”. The full wording has been implemented in the manuscript. LC2 and Axonal are the companies which took care of subjects’ randomization in the two original RCTs. LS-Means stands for “Least Squares Means”. The full wording has been introduced in the manuscript.

The tables are not in the correct order in the article

We agree. We propose to move Table 3 to results section “Speed of FMP’s effect on abdominal discomfort”. We hope that the flow is now clearer to the reader.

Round  2

Reviewer 1 Report

all comments have been addressed